# Changes in Emergency Department Case Severity and Length of Stay before and after COVID-19 Outbreak in Korea

**DOI:** 10.3390/healthcare10081540

**Published:** 2022-08-14

**Authors:** Yong-Seok Kim, In-Byung Kim, Seon-Rye Kim, Byung-Jun Cho

**Affiliations:** 1Department of Emergency Medical Service, College of Medical Science, Konyang University, 158, Gwanjeodong-ro, Seo-gu, Daejeon 35365, Korea; 2Department of Emergency Medicine, Myongji Hospital, Goyang, 24 Hwajung-dong, Deokyang-gu, Goyang-si 41227, Gyeonggi-do, Korea; 3Department of Healthcare Management, College of Health Science, Youngsan University, 258 Junam-ro, Yangsan-si 48015, Kyungsangnam-do, Korea; 4Department of Emergency Medical Technology, College of Health Science, Kangwon National University, 346 Hwangjo-gil, Dogye-up, Samcheok-si 25945, Gangwon-do, Korea

**Keywords:** COVID-19, emergency medical system, emergency department length of stay, severity

## Abstract

Severe patients visited regional emergency centers more frequently during the COVID-19 period, and disposition status warranting admission to the intensive care unit or resulting in death was more common during the COVID-19 period. This study was conducted to compare the characteristics and severity of patients, and emergency department length of stay before and after the COVID-19 outbreak. Subjects were 75,409 patients who visited the regional emergency medical center from 1 February 2019 to 19 January 2020 and from 1 February 2020 to 19 January 2021. Data was analyzed using the SPSS/WIN 22.0 program. The significance level was *p* < 0.05. The chi-square test and *t*-test were used for variables, and Cramer V was used for correlation. We found that the total number of patients visiting the emergency room decreased by 37.6% after COVID-19, but emergency department length of stay among severely ill patients increased by 203.7%. Additionally, the utilization rate of 119 ambulances and relatively more severe patients increased by 9.0% and by 2.1%. More studies about emergency department designs and operational programs should be conducted for better action not only during regular periods but also during periods of pandemic.

## 1. Introduction

The emergency medical system manages and operates all human and material resources essential for providing quality care at the state or local government level, including all matters related to first aid and care at hospitals [1]. The purpose of this emergency medical treatment is to reduce both mortality and human deterioration from disease by providing timely and safe care patients in moments of crisis.

In terms of emergency medical care, early recognition of changes in the epidemic of infectious diseases is important for preparation and response. The impact of infectious diseases on medical care varies depending on their characteristics. When severe acute respiratory syndrome became prevalent in Taiwan in 2003, emergency room use decreased due to fears of infection in hospitals [2]. The increase of infections at hospitals exacerbated the spread of the Middle East Respiratory Syndrome (MERS) epidemic in Korea in 2015, and MERS fears multiplied nationwide [3]. Patients were afraid to access medical services due to anxiety of a possible MERS infection at the hospital. This anxiety and accompanying fear eventually materialized as avoidance and reduction of medical care [4]. Since then, the need to improve the emergency medical system’s overcrowding and infectious disease management system has increased [5].

The recent 2019 SARS-CoV-2 (COVID-19) pandemic, which has been threatening human livelihood around the world, has caused other issues in terms of emergency medical services. Scholars such as Czeisler and Marymak have observed changes in delay or avoidance in access and use of emergency medical services as COVID-19 spread into communities [6]. These changes in the use of emergency medical services may increase the risk of morbidity and mortality with respect to patients’ treatment and health conditions [7]. In the U.S.A., 12.0% of adults reported that they delayed or avoided the use of medical services due to concerns related to COVID-19 [8]. Other studies have shown that cardiac arrest, stroke, and patient visits to emergency rooms have decreased since the COVID-19 pandemic [8,9]. Deaths have also increased due to the direct and indirect effects of COVID-19 [7]. Westgard further indicated that the number of patients visiting the emergency room after the COVID-19 pandemic decreased by 49.3% [10]. Naturally, even during the COVID-19 pandemic, individuals experiencing medical emergencies should be able to receive emergency medical treatment immediately, and urgent efforts are needed to ensure this [11].

In order to effectively respond to the outbreak of new infectious diseases, it may be important to predict changes in medical use according to the characteristics of existing infectious diseases and also to understand changes in medical use in various population groups. Moreover, understanding the changes in medical service use of severely ill emergency patients is an important factor in improving emergency medical response systems. Previous studies have shown that one COVID-19 strategy to prevent emergency room transmission [12,13] impacted the emergency medical system [14] and caused changes in related emergency room operations [15]. This study, thus, attempts to identify resulting changes that may have occurred in emergency rooms during the COVID-19 pandemic. In particular, this study examines the characteristics of patients using regional emergency medical centers, specifically focusing on changes in emergency room duration of stay and changes in emergency room use for severely ill patients.

## 2. Materials and Methods

### 2.1. Study Design and Subjects

Our research used data from a regional emergency medical center in a large Korean metropolitan area for a retrospective observation study. We used medical records of patients who visited their regional emergency medical center from 1 February 2019 to 19 January 2020 and from 1 February 2020 to 19 January 2021. This data excluded patients categorized at Level 5 (non-urgent) on Korean Triage and Acuity Scale (KTAS) [16], patients who visited for non-treatment purposes, and patients with incomplete medical records. The final analysis targets were 46,423 people before the outbreak of COVID-19 and 28,986 people afterwards. The study was conducted after deliberation and approval by the Institutional Research Ethics Committee of Myongji Hospital of the Myongji Medical Foundation (IRB File No. MJH 2021-08-031).

### 2.2. Korean Triage and Acuity Scale

The Korean Triage and Acuity Scale (KTAS) was introduced in Korea and has been applied to all emergency departments (Eds) since January 2016 [16]. Based on the Canadian Triage and Acuity Scale (CTAS), the KTAS was developed in accordance with the Korean medical environment. Using KTAS, medical personnel assess the severity of a patient’s condition according to five levels of decreasing urgency (level 1, resuscitation; level 2, emergent; level 3, urgent; level 4, less urgent; and level 5, non-urgent). The assessment is made based on the severity of symptoms, vital signs, and first and second considerations. The classification criteria for KTAS includes trauma, and the priority and extent of treatment for all ED patients, including trauma patients, are predicted by KTAS [17].

### 2.3. Statistical Analysis

The time periods before and after the outbreak of COVID-19 functioned as independent variables while gender, age, severity of stay, hospitalization type, hospitalization means, severe diseases, treatment results, and emergency room residence time were used as dependent variables. A chi-square test was performed for categorical variables, and a *t*-test was performed for continuous variables. A Cramer V was used for the relationship between severity of patients and severe diseases of patients visiting the emergency room before and after the COVID-19 outbreak. The SPSS/WIN 22.0 program was used for data analysis. Statistical significance level was *p* < 0.05.

## 3. Results

### 3.1. General Characteristics of Emergency Room Patients before and after the COVID-19 Outbreak

The total number of patients visiting the hospital after the outbreak of COVID-19 was 28,986, and this value was statistically significant and lower than the total number of patients visiting the hospital before COVID-19 at 46,423. (*p* < 0.001). There was no statistically significant difference in the gender ratio. The average age of patients visiting the hospital before and after the outbreak of COVID-19 was 46.84 (±25.47), which was statistically significant and higher than the average age of 36.94 (±27.77) before the outbreak of COVID-19 (*p* < 0.001) (Table 1).

### 3.2. Clinical Characteristics of Emergency Room Patients before and after the COVID-19 Outbreak

The severity ratio of patients visiting the emergency room before and after the outbreak of COVID-19 increased by 2.1% in the emergent patient group (KTAS 2), decreased by 2.2% in the urgent patient group (KTAS 3), and showed no difference in the less urgent patient group (KTAS 4). All values were statistically significant (*p* < 0.001). After COVID-19, the number of patients visiting the hospital and number of deaths on arrival decreased by 1.1% and 0.1%, respectively. Additionally, the number of patients visiting the hospital for reasons of trauma, drug intoxication, and self-harm increased by 0.8%, 0.1%, and 0.1%, respectively (*p* < 0.001). As for changes in emergency room patient transportation before and after COVID-19, the 119 (emergency hotline) ambulance ratio increased by 9.0% to the highest rate, 0.9% of private ambulances, 0.1% of others, and 10.1% of direct visits were statistically significant (*p* < 0.001) (Table 2).

### 3.3. The Emergency Department Length of Stay by Severity and Type of Diseases before and after COVID-19 Outbreak

The emergency department length of stay by severity in the periods before and after the COVID-19 outbreak significantly increased after the occurrence of COVID-19 in the emergent and urgent categories, or KTAS Levels 2 and 3 (*p* < 0.001). Level 4, which was less urgent, was not statistically significant (*p* = 0.171).

Data analysis also showed differences in emergency room stay time before and after COVID-19 across multiple severe diseases. These included the following: myocardial infarction from 74.58 (±82.51) minutes before to 213.97 (±263.44) minutes after; cerebral infarction from 178.28 (±80.09) minutes before to 377.12(±262.33) minutes after; severe trauma from 176.41 (±113.53) minutes before to 332.39 (±233.40) minutes after. In the specific case of post-resuscitation conditions, there was a statistically significant increase from 108.84 (±124.97) minutes before to 179.93 (±236.12) minutes after (*p* < 0.001) (Table 3, Figure 1 and Figure 2).

### 3.4. Medical Results of the Emergency Room Patients before and after the Outbreak of COVID-19

When comparing data for patients visiting the emergency room before and after the outbreak of COVID-19, it was discovered that the mortality rate increased by 0.7% after the outbreak, the intensive care unit admission rate increased by 1.5%, general hospitalization increased by 2.8%, and all patient item values decreased by 5.2%. All results were statistically significant (*p* < 0.001) (Table 4).

### 3.5. Correlation between Severe Diseases and Severity before and after the Outbreak of COVID-19

The correlation between severe diseases and severity of patients visiting the emergency room before and after the outbreak of COVID-19 was analyzed. The correlation coefficient between severe diseases and severity before the outbreak of COVID-19 was 0.391, and the correlation coefficient between severe diseases and severity after the outbreak of COVID-19 was 0.432, indicating a higher correlation after the outbreak of COVID-19 (Table 5 and Table 6).

## 4. Discussion

This study attempts to convey the necessary response capabilities of local emergency medical centers to combat a new infectious disease by identifying patient characteristics, condition severity, and duration of stay at local emergency medical centers during the COVID-19 pandemic.

After the outbreak of confirmed cases in Korea on 20 January 2020, there was a decrease in total visiting patients in both men and women, and there was a significant decrease in patients under the age of 19. In other age group except for age 19 and under, there were relative increases. One study about changes in emergency medical service use reported a 42% decrease in total emergency patients in the early stages of COVID-19 [6]. The decrease in the total number of patients visiting the emergency room and significant reduction in ages 19 and under continued during the pandemic.

One of the most interesting findings this study revealed is the decrease of children’s emergency rooms visitations. It is possible that the proportion of children under age 9 has decreased significantly because they have a relatively low severity compared to adults. It is believed that the fear of contracting COVID-19 had a larger impact than the tendency of parents to take their children to the emergency room due to worry, even though it is not an emergency (CITATION). A different study by Lazerini indicated a significant decrease of patients under the age of 17 who visited the pediatric emergency room [18], similar to this study’s results. Lazerini interpreted that the substantial decreases in children ED visitations in Italy might reflect a scarcity of available resources due to pandemic-related redistribution or reticence on the part of parents to risk exposure to COVID-19 in a health-care setting. These were in addition to considering lower rates of acute infections and trauma

Fear of infection is not limited to parental care, however. Recent studies have shown that in the early stages of the pandemic, patients from multiple age groups avoided emergency room use for fear of infection [18,19]. Patients suffering from acute or chronic diseases and who also avoid using emergency rooms due to vague fears of COVID-19 may worsen their extant conditions, and their resulting mortality rate may increase. [20]. In this study, it was also found that after the outbreak of COVID-19, the number of patients visiting regional emergency medical centers decreased significantly before the outbreak of COVID-19; however, compared to the decrease in visiting patients, the number and ratio of deceased patients were rather elevated, resulting in similar results to previous studies (CITATIONS). Even during the 2015 Korean MERS epidemic, the number of patients using emergency rooms decreased and mortality increased [4]. In this study, the mortality rate was higher than values during the MERS epidemic because the COVID-19 epidemic period was longer than MERS’s. Additionally, the MERS epidemic was resolved quickly, but the epidemic stage of COVID-19 is still ongoing.

After the COVID-19 pandemic, it can be seen that the average stay time in the emergency room increased significantly, compared to the period before. These results are also shown in previous studies [8,9,10,11,12,13,14,15], and there are possible explanations: it takes time to report patients’ COVID-19 results, the severity of patient illness for people who visit regional emergency medical centers has increased, and there has been a reduction of intensive care units for non-COVID-19 patients. Medical resources need to be managed in a way that the increase in stay time due to overcrowding in the emergency room increases the risk of exposure to infection to patients visiting the emergency room [1]. The average residence time of domestic emergency medical institutions before COVID-19 was 6.3 h, which is significantly different from the residence time of the top 20 emergency medical institutions before COVID-19, and the 14.5 h average [21]. In addition, the average stay time at a regional emergency medical center before COVID-19 was shorter than 4.19 h [22]. However, in previous studies, it was analyzed that the decrease in the number of intensive care unit beds increased the time of stay in the emergency room and thus was associated with an increase in the risk of death for patients with severe emergencies [23].

The time spent in the emergency room for patients increased for several diseases since the outbreak of COVID-19. In general, when the intensive care unit (ICU) is saturated, patients who are in urgent need of ICU treatment are first asked to transfer other medical institutions. Additionally, if resources are not available, they wait for hospitalization until an intensive care unit bed opens at the emergency medical center. According to recent statistics from the Central Emergency Medical Center, the number of critical patients who are transferred due to a lack of intensive care units has increased from 3069 in 2016 to 5087 in 2018 [24]. This is believed to be due to the intersection of challenges in accepting COVID-19 patients at other medical institutions. Additionally, the reduction of beds in the intensive care unit and the relocation of existing intensive care units and insufficient intensive care units has created more challenges. As Singer noted, increased residence time in the emergency room worsens the prognosis of critically ill patients; thus, it can be said that countermeasures are needed [25]. The increased length of stay of severe emergency patients leads to increased work for the emergency room medical staff, and while focusing on the treatment of new severe emergency patients, the concentration of existing hospitalized patients is reduced due to the nature of acute care [26].

During the COVID-19 pandemic, the severity ratio of patients visiting regional emergency medical centers increased in emergent (level 2) cases, decreased in urgent (level 3) cases, and showed no change in less-urgent (level 4) cases. In contrast, the decrease in the number of patients visiting the hospital was larger in emergency and non-emergency situations. Previous studies have revealed that the number of non-emergency patients visiting the hospital decreased significantly during the MERS epidemic [27]. A similar trend can be seen during the COVID-19 epidemic. In a study on the effect of COVID-19 on emergency medical care in Canada, 66.3% of patients visited the hospital compared to the rate of patients in the emergency room before COVID-19 occurred [28]. Such results seem to demonstrate that patient emergency room utilization rate is not greatly affected by the epidemic of infectious diseases.

In the present study, the mortality rate was 0.7%, the intensive care unit admission rate was 1.5%, and the discharge rate was −5.2% before the outbreak of COVID-19. According to the results of previous domestic studies, non-emergency medical centers showed an increase in the hospitalization rate of non-emergency patients, a decrease in the hospitalization rate of severe emergency patients, an increase in intensive care units, and a decrease in emergency room discharge rate [29,30,31]. These previous studies suggest that, eventually, the role and validity of each type of emergency medical institution could be revealed during COVID-19 epidemic. However, this study also demonstrated that the rate of deaths which increased before the COVID-19 outbreak had a partial effect on the increased stay time in the emergency room. In other words, in order to reduce the mortality rate for severely ill patients in the emergency room, it will be necessary to increase the rotation rate of intensive care beds by adjusting the hospitalization period of intensive care beds.

## 5. Conclusions

The results of this study showed a 37.6% decrease in total emergency room visits, a 203.7% increased emergency department length of stay for severely ill patients, a 9.0% increase in emergency hotline ambulance utilization rate for emergency patients, a 2.1% increased severity in the emergency room, and a two-fold increase in deceased patients after the COVID-19 outbreak.

In conclusion, emergency room deaths of severely ill patients increased after the outbreak of COVID-19, and it would be difficult to maintain continuous emergency medical capabilities for severely ill patients whose rising number continues to impact the workload of medical personnel. Therefore, it is necessary to devise strategies that will secure appropriate medical personnel and the appropriate number of beds. Dong this will allow for enhanced infection control, redesigned emergency room spaces, and improved management of an increasing number of severely ill patients.

## Figures and Tables

**Figure 1 healthcare-10-01540-f001:**
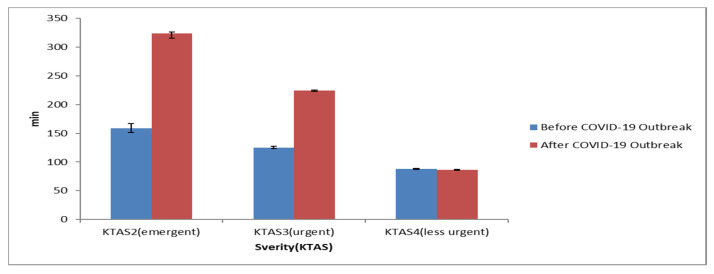
The difference in the emergency department length of stay (EDLOS) by severity before and after COVID-19 outbreak.

**Figure 2 healthcare-10-01540-f002:**
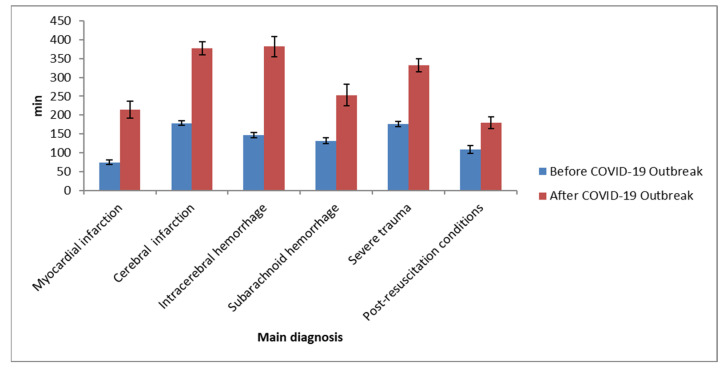
The difference in the emergency department length of stay (EDLOS) by main diagnosis before and after COVID-19 outbreak.

**Table 1 healthcare-10-01540-t001:** General characteristics of emergency room patients before and after the COVID-19 outbreak.

Characteristics	Item	Before COVID-19 Outbreak *n* (%) or M (±SD)	After COVID-19 Outbreak *n* (%) or M (±SD)	*p*
Total		46,423	28,986	<0.001
Gender	Male	23,340 (50.3%)	14,707 (50.7%)	0.218
	Female	23,083 (47.9%)	14,279 (49.3%)	
Age (years)	0–9	12,481 (26.9%)	3026 (10.4%)	<0.001
	10–19	3499 (7.5%)	1784 (6.2%)	
	20–29	4375 (9.4%)	3616 (12.5%)	
	30–39	4199 (9.0%)	3006 (10.4%)	
	40–49	4726 (10.2%)	3338 (11.5%)	
	50–59	5806 (12.5%)	4345 (15.0%)	
	60–69	4105 (8.8%)	3353 (11.6%)	
	70–79	3543 (7.6%)	2952 (10.2%)	
	>= 80	3689 (7.9%)	3566 (12.3%)	
	Mean (±SD)	36.94 (±27.77)	46.84 (±25.47)	<0.001

**Table 2 healthcare-10-01540-t002:** Clinical characteristics of emergency room patients before and after the COVID-19 outbreak.

Characteristics	Item	Before COVID-19 Outbreak *n* (%)	After COVID-19 Outbreak *n* (%)	*p*
Total		46,423	28,986	<0.001
Severity (KTAS)	Level 2 (emergent)	2510 (5.4%)	2176 (7.5%)	<0.001
	Level 3 (urgent)	22,021 (47.4%)	13,116 (45.2%)	
	Level 4 (less urgent)	21,892 (47.2%)	13,694 (47.2%)	
Reasons of	Disease	33,678 (72.6%)	20,700 (71.5%)	<0.001
visiting	Trauma	12,194 (26.3%)	7851 (27.1%)	
The hospital	DI	212 (0.5%)	175 (0.6%)	
	Suicide	117 (0.3%)	130 (0.4%)	
	DOA	170 (0.4%)	84 (0.3%)	
Means of visiting	119 Ambulance	9573 (20.6%)	8570 (29.6%)	<0.001
The hospital	Private Ambulance	1929 (4.2%)	1484 (5.1%)	
	By oneself	34,890 (75.2%)	18,864 (65.1%)	
	Others	31 (0.1%)	68 (0.2%)	

KTAS (The Korean Triage and Acuity Scale), DI (drug intoxication), DOA (dead on arrival).

**Table 3 healthcare-10-01540-t003:** The difference in the emergency department length of stay (EDLOS) by severity before and after COVID-19 outbreak.

Characteristics	Item	Before COVID-19 Outbreak M ± SD (min)	After COVID-19 Outbreak M ± SD (min)	*p*
Severity (KTAS)	Level 2 (emergent)	158.85 ± 123.01 (*n* = 2510)	323.64 ± 358.33 (*n* = 2176)	<0.001
	Level 3 (urgent)	125.10 ± 97.60 (*n* = 22,021)	224.73 ± 228.08 (*n* = 13,116)	<0.001
	Level 4 (less urgent)	87.63 ± 71.99 (*n* = 21,892)	86.47 ± 86.20 (*n* = 13,694)	0.171
Main	Myocardial infarction	74.58 ± 82.51 (*n* = 159)	213.97 ± 263.44 (*n* = 131)	<0.001
diagnosis	Cerebral infarction	178.28 ± 80.09 (*n* = 155)	377.12 ± 262.23 (*n* = 231)	<0.001
	Intracerebral hemorrhage	146.62 ± 83.63 (*n* = 169)	381.75 ± 353.62 (*n* = 167)	<0.001
	Subarachnoid hemorrhage	131.49 ± 52.62 (*n* = 47)	252.72 ± 209.84 (*n* = 53)	<0.001
	Severe trauma	176.41 ± 113.53 (*n* = 237)	332.39 ± 233.40 (*n* = 180)	<0.001
	Post-resuscitation conditions	108.84 ± 124.97 (*n* = 155)	179.93 ± 263.12 (*n* = 232)	0.001

KTAS (The Korean Triage and Acuity Scale).

**Table 4 healthcare-10-01540-t004:** Medical results of patients visiting the emergency room before and after the outbreak of COVID-19.

Characteristics	Item	Before COVID-19 Outbreak (*n* = 46,423)	After COVID-19 Outbreak (*n* = 28,986)	*p*
Medical	Death	318 (0.7%)	409 (1.4%)	<0.001
Results	Intensive care unit admission	2127 (4.6%)	1763 (6.1%)	
	General hospitalization	7831 (16.9%)	5700 (19.7%)	
	Transfer	190 (0.4%)	159 (0.5%)	
	Discharge	35,957 (77.5%)	20,955 (72.3%)	

**Table 5 healthcare-10-01540-t005:** Correlation between severe diseases and severity before the outbreak of COVID-19.

Severe Disease	Total	Severity (KTAS)			Cramer V	95%CI
		Level 2 (Emergent)	Level 3 (Urgent)	Level 4 (Less Urgent)		
Myocardial infarction	159 (17.2%)	123 (23.1%)	36 (9.8%)	0 (0.0%)	0.391	0.361–0.430
Cerebral infarction	155 (16.8%)	41 (7.7%)	111 (30.2%)	3 (13.6%)		
Intracerebral hemorrhage	169 (18.3%)	103 (19.3%)	65 (17.7%)	1 (4.5%)		
Subarachnoid hemorrhage	47 (5.1%)	31 (5.8%)	16 (4.4%)	0 (0.0%)		
Severe trauma	237 (25.7%)	80 (15.0%)	139 (37.9%)	18 (81.8%)		

KTAS (The Korean Triage and Acuity Scale), CI (Confidence Interval).

**Table 6 healthcare-10-01540-t006:** Correlation between severe diseases and severity after the outbreak of COVID-19.

Severe Disease	Total	Severity (KTAS)			Cramer V	95%CI
		Level 2 (Emergent)	Level 3 (Urgent)	Level 4 (Less Urgent)		
Myocardial infarction	131 (13.2%)	109 (18.2%)	22 (5.8%)	0 (0.0%)	0.432	0.404–0.470
Cerebral infarction	231 (23.2%)	64 (10.7%)	167 (43.9%)	0 (0.0%)		
Intracerebral hemorrhage	167 (16.8%)	98 (16.4%)	68 (17.9%)	1 (6.7%)		
Subarachnoid hemorrhage	53 (5.3%)	33 (5.5%)	20 (5.3%)	0 (0.0%)		
Severe trauma	180 (18.1%)	65 (10.9%)	101 (26.6%)	14 (93.3%)		

KTAS (The Korean Triage and Acuity Scale), CI (Confidence Interval).

## Data Availability

The study did not report any data.

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
