# Peer review of "Changes in Emergency Department Case Severity and Length of Stay before and after COVID-19 Outbreak in Korea"

_healthcare, 2022, doi:10.3390/healthcare10081540_

Round 1
Reviewer 1 Report
A study to analyze the changes in the severity and emergency department length of stay before and after the COVID-19 outbreak in Korea is presented in this manuscript. The manuscript is well written the following are my suggestions.
In the abstract, the decrease and increase statements can be supported with the % values.
Box plots can be used to compare the Pre and Post covid patients admitted for different reasons.
Graphs can be plotted for the length of stay during pre and post-covid.
Conclusion must be supported by numeric results.
Author Response
A study to analyze the changes in the severity and emergency department length of stay before and after the COVID-19 outbreak in Korea is presented in this manuscript. The manuscript is well written the following are my suggestions.
In the abstract, the decrease and increase statements can be supported with the % values.
- I added the % values to statements that mentioned values increasing and decreasing.
Box plots can be used to compare the Pre and Post covid patients admitted for different reasons.
- I added graphs to compare values for pre- and post-COVID patients.
Graphs can be plotted for the length of stay during pre and post-covid.
- I added graphs for the length of stay during pre- and post-COVID periods.
Conclusion must be supported by numeric results.
- I edited the conclusion to include numeric results.
Reviewer 2 Report
Intro, background and methods are fine.
The main area that needs revision is the discussion.
Most of the first third of the discussion reviews detailed data not presented in the paper about monthly utilization and other studies in the area. It is distracting and not really needed.
The most interesting findings should be highlighted.
- Utilization was down - on average
- Reductions varied by age and severity
- Biggest reduction is for kids.
Need to be more direct/explicit --What is important about these findings and why.
For example, it looks like patients did a good job of self triaging. Most of the reduction is in lower severity categories. Does this mean we can train patients to use alternatives to hospital care in the future?
Finally might include that these findings for Korea match what has been seen in other countries.
Author Response
Intro, background and methods are fine.
- Thank you.
The main area that needs revision is the discussion.
- I have edited the discussion according to your comments.
Most of the first third of the discussion reviews detailed data not presented in the paper about monthly utilization and other studies in the area. It is distracting and not really needed.
=> I deleted any discussion parts that were distracting to the reader.
The most interesting findings should be highlighted.
- Utilization was down - on average
- Reductions varied by age and severity
- Biggest reduction is for kids.
=> I did emphasize those finding.
One of the most interesting findings this study revealed is the decrease of children’s emergency rooms visitations. It is possible that the proportion of children under age 9 has decreased significantly because they have a relatively low severity compared to adults. It is believed that the fear of contracting COVID-19 had a larger impact than the tendency of parents to take their children to the emergency room due to worry, even though it is not an emergency (CITATION). A different study by Lazerini indicated a significant decrease of patients under the age of 17 who visited the pediatric emergency room [18], similar to this study's results. Lazerini interpreted that the substantial decreases in children ED visitations in Italy might reflect a scarcity of available resources due to pandemic-related redistribution or reticence on the part of parentsto risk exposure to COVID-19 in a health-care setting. These were in addition to considering lower rates of acute infections and trauma.
Need to be more direct/explicit --What is important about these findings and why.
=> The proportion of children under the age of 9 has decreased significantly because children have a relatively low severity compared to adults. Even though children may not have a medical emergency per se, parents have a tendency to take their children to the emergency room because the parents are worried. But nowadays the fear of getting sick with COVID-19 has played a greater role than the tendency of parents to take their children to the emergency room. Thus, it is believed that most of the reduction is in lower severity categories.
For example, it looks like patients did a good job of self-triaging. Most of the reduction is in lower severity categories. Does this mean we can train patients to use alternatives to hospital care in the future?
=> Most of the reduction is in lower severity categories.
Finally might include that these findings for Korea match what has been seen in other countries.
=> A different study by Lazerini (2020) showed a significant decrease of patients under the age of 17 who visited the pediatric emergency rooms in Italy. The results are similar to this study's results. Lazerini interpreted that the substantial decreases in children emergency room visitations might reflect a scarcity of available resources due to pandemic-related redistribution or to reticence on the part of parents and caregivers to risk exposure to COVID-19 in a health-care setting. These would be in addition to lower rates of acute infections and trauma.
Reviewer 3 Report
The proposed study, despite being well written and conducted (even statistically) and endowed with a clear design, reaches fairly obvious conclusions and adds nothing new to the current knowledge available.
Author Response
The proposed study, despite being well written and conducted (even statistically) and endowed with a clear design, reaches fairly obvious conclusions and adds nothing new to the current knowledge available.
=>Thank you for your comments.
Our hope is that this study will expand the current information ecosystem specifically with regards to COVID-19.
To begin, the results of this study are different from previous studies. This study dealt with the high severity, by evaluating emergency room stay time by severity and emergency room stay time by severe disease after the outbreak of COVID-19. Previous studies (7,9,20,28) have mainly dealt with issues such as a decrease in emergency room use according to general characteristics, a decrease in emergency room patient influx by disease, and a change in the distribution of emergency room visits by severity. This study is different, however. We focus on the severity. The increase in emergency room time by severity occurs mainly in patients with conditions of high severity. So, it is considered that there is a limit to providing optimal medical services in the limited environment of the emergency room. In the end, the increased stay time in the emergency room of severely ill patients is expected to fall short of the medical level provided by specialized intensive care units or specialized wards. Such constraints may worsen the patient's condition. In addition, for newly introduced severely ill patients, emergency room medical staff might delay the appropriate time for first aid due to an environment in which they cannot concentrate. Therefore, measures are needed to reduce the time patients spent in the emergency room to maintain emergency room function in pandemic situations (e.g. COVID-19).
As this study is tied to the historical event of COVID-19, the value of its findings speak directly to emergency room operations management during times of a public health crisis.
Round 2
Reviewer 3 Report
I want to thank the kind authors for their reply, which cointain a very clear and interesting explanation. I have seen that all the observations/proposals of the other reviewers are being satisfied, so I go in the same direction and I do not oppose publication this time.